# Metformin in Esophageal Carcinoma: Exploring Molecular Mechanisms and Therapeutic Insights

**DOI:** 10.3390/ijms25052978

**Published:** 2024-03-04

**Authors:** Stavros P. Papadakos, Alexandra Argyrou, Vasileios Lekakis, Konstantinos Arvanitakis, Polyxeni Kalisperati, Ioanna E. Stergiou, Ippokratis Konstantinidis, Dimitrios Schizas, Theocharis Koufakis, Georgios Germanidis, Stamatios Theocharis

**Affiliations:** 1First Department of Pathology, Medical School, National and Kapodistrian University of Athens, 75 Mikras Asias Street, Goudi, 11527 Athens, Greece; stpap@med.uoa.gr; 2Academic Department of Gastroenterology, Laikon General Hospital, Athens University Medical School, 11527 Athens, Greece; argyalex89@gmail.com (A.A.); lekakisvas@gmail.com (V.L.); 3First Department of Internal Medicine, AHEPA University Hospital, Aristotle University of Thessaloniki, 54636 Thessaloniki, Greece; arvanitak@auth.gr; 4Basic and Translational Research Unit (BTRU), Special Unit for Biomedical Research and Education (BRESU), Faculty of Health Sciences, School of Medicine, Aristotle University of Thessaloniki, 54636 Thessaloniki, Greece; 5Pathophysiology Department, School of Medicine, National and Kapodistrian University of Athens, 11527 Athens, Greece; xkalisperati@med.uoa.gr (P.K.); stergiouioa@med.uoa.gr (I.E.S.); 6Department of Internal Medicine, University of Connecticut, Farmington, CT 06030, USA; konstantinidis@uchc.edu; 7First Department of Surgery, Laikon General Hospital, National and Kapodistrian University of Athens, 11527 Athens, Greece; schizasad@gmail.com; 8Second Propaedeutic Department of Internal Medicine, General Hospital “Hippokration”, Aristotle University of Thessaloniki, 54642 Thessaloniki, Greece; thkoyfak@hotmail.com

**Keywords:** esophageal cancer (EC), esophageal adenocarcinoma (EAC), esophageal squamous cell carcinoma (ESCC), metformin, immunotherapy

## Abstract

Esophageal cancer (EC) remains a formidable malignancy with limited treatment options and high mortality rates, necessitating the exploration of innovative therapeutic avenues. Through a systematic analysis of a multitude of studies, we synthesize the diverse findings related to metformin’s influence on EC. This review comprehensively elucidates the intricate metabolic pathways and molecular mechanisms through which metformin may exert its anti-cancer effects. Key focus areas include its impact on insulin signaling, AMP-activated protein kinase (AMPK) activation, and the mTOR pathway, which collectively contribute to its role in mitigating esophageal cancer progression. This review critically examines the body of clinical and preclinical evidence surrounding the potential role of metformin, a widely prescribed anti-diabetic medication, in EC management. Our examination extends to the modulation of inflammation, oxidative stress and angiogenesis, revealing metformin’s potential as a metabolic intervention in esophageal cancer pathogenesis. By consolidating epidemiological and clinical data, we assess the evidence that supports metformin’s candidacy as an adjuvant therapy for esophageal cancer. By summarizing clinical and preclinical findings, our review aims to enhance our understanding of metformin’s role in EC management, potentially improving patient care and outcomes.

## 1. Introduction

### 1.1. Exploring Esophageal Cancer: Trends, Treatments, and Future Horizons

Esophageal cancer (EC), ranking sixth in global cancer mortality, comprises two distinct diseases: esophageal adenocarcinoma (EAC) and esophageal squamous cell carcinoma (ESCC) [1,2]. EAC is linked to Barrett’s esophagus (BE), influenced by factors like gastroesophageal reflux disease (GERD) and obesity, while ESSC arises from squamous cells, with tobacco and alcohol as notable risks [3]. The Global Burden of Diseases Study 2017 analyzed data revealing a 22% decline in global incidence rates from 1990 to 2017, except in North America and western sub-Saharan Africa. Incidence trends correlate with socioeconomic development [4]. Over the past three decades, the landscape of EC has undergone significant changes in histology and anatomical distribution. Initially dominated by ESSC, the incidence of EC in the United States has seen a decline since the 1970s, with EAC emerging as the predominant histology in the early 1990s [5]. Despite the challenges in treating advanced-stage EC with resistant biological features, substantial progress has been achieved in curative approaches [6]. Ongoing randomized controlled trials (RCTs), advancements in endotherapy, staging, surgery, and scientific analysis, coupled with an enhanced understanding of genomics and the tumor microenvironment (TME), hold promise for further discoveries and improved cure rates [6]. The curative model for locally advanced EC is expected to evolve soon, incorporating information on HER2 status, PD-L1 expression, MSI, and overall mutational burden [7]. Biomarkers such as circulating tumor DNA show promise [8]. Despite advancements, there is a need for new therapeutic options in managing advance EC. Emerging evidence from other malignancies suggest that repurposing of metformin could be a promising option [9].

### 1.2. Understanding the Mechanisms and Pathophysiology of Esophageal Carcinoma

While the analytic presentation of these mechanisms has been addressed elsewhere [1], our manuscript focuses on summarizing the fundamental aspects, recognizing that a comprehensive analysis extends beyond the scope of our work.

As regards the pathogenesis of ESCC, it involves the progression from basal cell hyperplasia and dysplasia to carcinoma in situ, with dysregulation of TP53 and cell cycle regulators as prominent features [10]. Despite challenges in accurately stratifying risk based on differentiation between normal and dysplastic tissue, potential biomarkers like TNFAIP3 and CHN have been identified [10]. Large-scale genome-wide association studies (GWAS) in China have identified susceptibility loci for ESCC, including 10q23 (PLCE1), 5q31.2 (TMEM173), 17p13.1 (ATP1B2 near TP53), and, specific to high-risk areas, the HLA class II region (6p21.32) [11]. Genetic variability in detoxification processes, exemplified by functional variants in alcohol dehydrogenase IB (ADH1B) and aldehyde dehydrogenase 2 (ALDH2), interacts with lifestyle factors to heighten ESCC risk in the Japanese population [12]. Recent large-scale sequencing studies have shed light on the mutational landscape of invasive ESCC, emphasizing dysregulated pathways such as cell cycle regulators, tyrosine kinase receptors, chromatin remodeling, and embryonic pathways [13]. The TCGA dataset highlighted prevalent mutations in TP53, MLL2, and NFE2L2, along with amplifications in SOX2/TP63 and FGFR1 [14]. Notably, the EGFR signaling pathway and PIK3CA were activated in a significant percentage of tumors, presenting potential therapeutic targets with tyrosine kinase inhibitors, a strategy successfully employed in other solid tumors.

The pathogenesis of EAC involves reflux-induced DNA damage, especially A>C transversions, contributing to BE development as an adaptive response to squamous mucosa injury. Barrett’s mucosa, a pre-neoplastic tissue, frequently contains somatic genetic alterations, including TP53 and SMAD4 mutations, contributing to carcinogenesis [15]. EAC development involves chromosomal instability, copy number alterations, and mutations in genes such as TP53, CDK2NA, and ARID1A. The heterogeneity of EAC poses challenges for targeted therapies, and whole-genome sequencing has identified subgroups with potential therapeutic implications, such as DNA damage repair-deficient subtypes benefiting from PARP/ATR inhibitors or platinum-based chemotherapy and subgroups with a high mutational burden responding to immuno-oncology therapies [16]. Helicobacter pylori infection shows an inverse association with BE/EAC risk, and improved socioeconomic conditions leading to decreased Helicobacter pylori seropositivity may contribute to rising EAC rates [17]. Host genetics contribute to up to one-third of sporadic BE and EAC risk, with certain genetic loci identified through GWAS [1].

### 1.3. Metformin: Unraveling Physiological Functions and Anti-Cancer Mechanisms

The intricate mechanisms of metformin are extensively examined elsewhere [18], and their detailed presentation goes beyond the scope of our manuscript. Metformin primarily acts to enhance insulin sensitivity and reduce hepatic glucose output, resulting in lowered insulin and glucose levels [19]. Proposed mechanisms include increased insulin receptor expression, modulation of the incretin pathway, and improved sensitivity due to enhanced tyrosine kinase activity [18]. Metformin is suggested to augment glucagon-like peptide-1 (GLP-1) secretion and beta-cell GLP-1 receptor expression, possibly via PPAR-a [20]. The anti-hyperglycemic effect is attributed to decreased hepatic gluconeogenesis, potentially through reduced uptake of gluconeogenic substrates or inhibition of related enzymes. Metformin, preferentially taken up by the liver by organic cation transporter 1 (OCT1), interacts with the mitochondrial membrane, inhibiting the electron transport system [21]. This leads to a rise in the AMP/ATP ratio, activating AMP-activated protein kinase (AMPK), which, in turn, shifts the cell to a catabolic state, promoting cellular energy balance by inhibiting protein, lipid, and glucose synthesis while increasing glucose and fatty acid uptake [22]. Although metformin does not directly activate AMPK, its impact on the mitochondria, particularly the respiratory complex 1, is considered pivotal in this activation process [23].

Metformin inhibits cancer through direct and indirect pathways, impacting multiple molecular mechanisms [18]. The activation of AMPK plays a crucial role, inhibiting cellular proliferation and inducing G1-phase cell cycle arrest [24]. Metformin also activates the multidrug resistance 1 (MDR1) gene, potentially reducing drug failure in chemotherapy [25]. Additionally, it inhibits lipogenic enzymes, leading to reduced fatty acid availability for tumor cells. The inhibition of the mammalian target of rapamycin (mTOR) pathway by metformin suppresses cancer progression and angiogenesis [26]. Metformin influences insulin and insulin-like growth factors (IGFs), reducing plasma insulin and IGF-1 levels, known mitogen for cancer cells [27]. Other signaling pathways affected include Ras/Raf/mitogen-activated protein kinase (MAPK) [28], nuclear factor kappa B (NF-kB) [29], and human epidermal growth factor receptor-2 (HER2) [30]. Metformin exhibits anti-angiogenic effects by attenuating pro-angiogenic inflammatory stimuli [31]. It may inhibit cancer stem cells, improve chemotherapy response, stimulate cytotoxic T lymphocytes, and suppress DNA damage, contributing to its overall anti-cancer properties [18]. The above are illustrated in brief in Figure 1.

## 2. Metformin in Esophageal Cancer: Exploring Its Clinical Significance and Therapeutic Implications

The role of metformin in the context of EC is currently under in-depth clinical investigation, aimed at unraveling its significance and therapeutic potential. Research on metformin’s role in esophageal cancer has yielded mixed results, revealing a complex connection. Some studies suggest a potential reduction in cancer risk and improved efficacy in anti-cancer treatments. Ongoing clinical investigations are crucial for elucidating the role of metformin in EC [32]. However, the existing body of evidence remains inconclusive, emphasizing the need for further comprehensive research to establish the precise clinical significance of metformin.

### 2.1. Metformin for Esophageal Cancer Risk Reduction

Lee et al. utilized data from the Taiwanese National Health Insurance (NHI) organization to conduct a prospective cohort analysis involving 800,000 individuals [33]. They found that metformin use was associated with a reduced risk of esophageal cancer development. This protective effect remained significant after adjusting for various factors, including age, gender, comorbidity score, duration of metformin use, and the use of other anti-hyperglycemic medications. Furthermore, the study examined different doses of metformin and observed gender-specific effects. Female metformin users had a significantly lower risk of EC, while the risk reduction in male users was not statistically significant. In summary, the findings suggest a potential protective effect of metformin against EC, emphasizing its role in reducing the risk of this specific type of cancer [33]. Tseng et al. investigated the impact of metformin on EC risk among Taiwanese patients with type 2 diabetes mellitus (T2DM). They documented that the incidence of EC was significantly lower in metformin users (25.03 per 100,000 person-years) compared to never users (50.87 per 100,000 person-years), with an overall hazard ratio (HR) of 0.487 (95% confidence intervals: 0.347–0.684). The HR based on cumulative duration of metformin use demonstrated a decreasing trend, suggesting a protective effect with longer use [34]. Becker et al. conducted a case-control analysis investigating the relationship between the use of metformin and other anti-diabetic drugs and the risk of EC. They used data from the UK-based General Practice Research Database (GPRD) and identified cases of individuals aged 40–89 years who were diagnosed with esophageal cancer between 1994 and 2010, and selected ten controls for each case. The controls were matched based on age, sex, calendar time, and the number of years of active history. They found that long-term use (over 30 prescriptions) of metformin did not show a significant association with an altered risk of EC, with an adjusted OR of 1.23 and a 95% CI of 0.92–1.65 [35]. Wang et al. aimed to investigate the relationship between metformin use and the risk of developing ESCC. They conducted a population-based cohort study in Sweden from 2005 to 2015, involving 8.4 million participants. Among them, 411,603 were metformin users, and they were compared to 4,116,030 nonusers. They found that metformin users had a decreased risk of ESCC compared to nonusers, with a more significant reduction in risk among new metformin users and individuals aged 60–69 years. This suggests that metformin may have a protective effect against the development of ESCC [36]. Finally, Loomans-Kropp et al. investigated the impact of common drugs, including metformin, on reducing the risk of EAC. They suggested that the use of proton pump inhibitors (PPIs), statins, non-steroidal anti-inflammatory (NSAIDs) drugs, or metformin may reduce the risk of EAC. Metformin use was associated with reduced odds of EAC, with an odds ratio (OR) of 0.76 (95% Cl, 0.62–0.93). This indicates that metformin was associated with a 24% reduction in the odds of developing EAC [37].

Data from two recent meta-analyses present contradictory findings [38,39]. Chen et al. examined the association between metformin use and the risk of EC. The study included seven research papers with a total of 5,426,343 subjects. Their findings indicate that metformin use is associated with a reduced risk of OC, with a pooled HR of 0.69 and a 95% confidence interval (CI) of 0.54 to 0.87 (*p* < 0.001), suggesting that metformin may have a protective effect against EC, emphasizing the need for further well-designed studies to provide additional insights into this association [38]. Conversely, Wu et al. assessed the effect of metformin on esophageal cancer risk in patients with T2DM through a systematic review and meta-analysis. They indicated that metformin did not significantly reduce the risk of EC in patients with T2DM (HR 0.88, 95% CI 0.60–1.28, *p* > 0.05). However, subgroup analyses by geographic location revealed a significant reduction in esophageal cancer risk associated with metformin in Asian patients with T2DM (HR 0.59, 95% CI 0.39–0.91, *p* = 0.02), with no heterogeneity between studies [39]. In conclusion, while metformin did not show a notable reduction in EC risk in T2DM patients overall, a significant risk reduction was observed in Asian populations, although further clarification is needed. The above are summarized in Table 1.

### 2.2. Metformin’s Impact on Esophageal Cancer Survival

Wang et al. investigated the relationship between diabetes, metformin use, and survival in EC patients focusing on all-cause and disease-specific mortality [40]. They suggested that EC patients with diabetes but not using metformin had increased all-cause mortality. In contrast, non-diabetic patients and diabetic patients using metformin showed decreased all-cause mortality. They also found a trend of decreasing all-cause mortality with a higher daily dose of metformin. They did not find associations between mortality outcomes and other antidiabetic medications like sulfonylureas, insulin, or thiazolidinedione [40]. However, more research is needed to determine the specific impact of metformin on survival in EC. Skinner et al. focused on the impact of metformin use on the response to therapy in EAC patients undergoing neoadjuvant chemoradiation [41]. They analyzed data from 285 patients treated with concurrent chemoradiation followed by esophagectomy. Among them, 29 were diabetic and taking metformin, 21 were diabetic but not taking metformin, and 235 were non-diabetic. They found that the pathologic complete response (CR) rate was higher in patients taking metformin (34.5%) compared to diabetic patients not taking metformin (4.8%) and non-diabetic patients (19.6%). The higher metformin dose was associated with a greater CR rate. Metformin use was independently associated with pathologic CR, and it was also linked to decreased loco-regional failure following radiation. The findings suggest that metformin may enhance the response to chemoradiation therapy in esophageal cancer, with a dose-dependent effect [41]. Spierings et al. aimed to explore the impact of metformin use on pathological response, overall survival, and disease-free survival in patients with resectable esophageal cancer undergoing neoadjuvant chemo(radio)therapy with curative intent [42]. The research included 461 patients who underwent esophagectomy between March 1994 and September 2013. Among the patients, 43 had T2DM, with 32 using metformin. The findings revealed that metformin use did not lead to higher pathological response rates compared to non-metformin users. They suggested that, contrary to findings in other tumor types, metformin may not have a beneficial effect on EC [42]. Van De Voorde et al. delved into the potential benefits of metformin in patients treated for EC [43]. They included 196 patients categorized as non-diabetic, diabetic and not taking metformin, or diabetic and taking metformin. Most patients underwent trimodality therapy (surgery, chemotherapy and radiation therapy). They found an overall pathologic CR rate of 26%, with 25% for non-metformin users and 39% for diabetics taking metformin. The two-year OS rate was 59%, and metformin use was associated with significantly better distant metastasis-free survival and OS rates. Multivariate analysis confirmed that metformin treatment significantly prolonged survival. They concluded that, in their population-based investigation, metformin use was linked to improved overall and distant metastasis-free survival in patients with EC [43].

A recent meta-analysis provides a wealth of evidence towards this direction [44]. Sakamoto et al. presented the first meta-analysis investigating the impact of metformin on neoadjuvant chemoradiotherapy (NACRT) in rectal and esophageal/gastroesophageal cancer patients. They reported that the metformin group exhibited an increased pathologic CR rate compared to the non-metformin group. Notably, diabetic patients, who typically face a poorer cancer prognosis, demonstrated an association between metformin use and the pCR rate. The study focused on advanced cancers of grade T3 or higher, with advanced cancers contributing significantly to the observed association between metformin and the pCR rate. The study suggested that metformin’s effectiveness may be particularly pronounced in EAC, as no effect was demonstrated in studies including patients with ESCC. The anti-cancer effects of metformin are attributed to mechanisms such as mTOR inhibition and synergistic effects with chemotherapy and radiotherapy. Further research, including randomized controlled trials, is encouraged to elucidate metformin’s efficacy, especially in non-diabetic patients. These are briefly mentioned in Table 2.

### 2.3. Metformin as a Chemopreventive Agent

An accumulating body of evidence strongly suggests that metformin holds promise as a chemopreventive agent. Arai et al. conducted a multicenter retrospective cohort analysis to investigate the chemopreventive effects of commonly used drugs on ESCC and EAC [45]. They showed that the use of PPIs, aspirin, cyclooxygenase-2 inhibitor (COX2I), steroids, statins, and metformin was associated with a lower risk of ESCC compared to non-use. Specifically, the adjusted ORs (aORs) for ESCC were 0.48 for PPIs, 0.32 for aspirin, 0.70 for COX2I, 0.19 for steroids, 0.43 for statins, and 0.42 for metformin. Contrarily, Chak et al. aimed to assess the potential chemopreventive effects of metformin on BE, focusing on its impact on phosphorylated S6 kinase (pS6K1), a biomarker of insulin pathway activation [46]. In a randomized, double-blind, placebo-controlled phase 2 trial with 74 BE subjects, metformin (daily up to 2000 mg for 12 weeks) did not significantly reduce esophageal pS6K1 levels compared to placebo. While metformin did show an almost significant reduction in serum insulin levels and insulin resistance, it did not affect cell proliferation or apoptosis in esophageal tissues. These findings do not support metformin as a chemopreventive agent for BE-associated carcinogenesis. In the same direction, Agrawal et al. aimed to explore the impact of metformin use on the risk of developing esophageal adenocarcinoma in patients with BE [47]. Over a 20-year period, 583 patients with BE or EAC were identified. Age, smoking, and diabetes mellitus were identified as significant risk factors for EC, while statin use showed a protective effect. However, metformin use did not exhibit a statistically significant association, suggesting it did not demonstrate a protective effect against the development of EAC in this study. Notably, Antonowicz et al. focused on the presence of volatile aldehydes in the breath of EAC patients and their potential for early diagnosis improvement [48]. They revealed that EAC patients exhibit an enrichment of volatile aldehydes, particularly short-chain alkanals and medium-chain alkanals, including decanal, in biopsies and adjacent tissues. The identified short-chain alkanals form DNA adducts, indicating genotoxicity and inadequate detoxification in EAC. They suggested that metformin plays a role in enhancing aldehyde detoxification, as evidenced by its ALDH-enhancing and aldehyde-scavenging effects. Aldehyde accumulation in EAC is associated with genotoxicity, and metformin’s potential to augment aldehyde detoxification may have implications for chemopreventative strategies in precancerous conditions like Barrett’s esophagus. Additionally, the findings underscore the clinical relevance of exhaled aldehydes as potential diagnostic biomarkers for early detection of EAC [48].

Summarizing, metformin’s role in EC presents conflicting results, with some studies suggesting a potential for risk reduction and enhanced anti-cancer treatment efficacy. However, the evidence remains inconclusive, warranting further research to determine its precise clinical significance. The above are briefed in Table 3.

## 3. Understanding Metformin’s Molecular Mechanisms in EC

Preclinical data indicates that metformin exhibits potential for both the prevention and treatment of EC. This potential arises from its ability to target various fundamental aspects of cancer biology, including proliferation and apoptosis, mitigation of drug resistance, autophagy, angiogenesis, metastasis, and epigenetic regulation. A detailed examination of these effects will be presented in the subsequent analysis.

### 3.1. The Regulatory Effects of Metformin on EC Cellular Proliferation

Over the past decade, numerous studies have explored the impact of the oral anti-hyperglycemic medication metformin on esophageal cancer cells. These investigations have consistently presented evidence of metformin’s dose-dependent inhibitory influence on cell proliferation and its ability to induce cell cycle arrest, both in cell cultures and in an animal model, by suppressing tumor growth.

Xu et al. demonstrated that metformin hinders EC proliferation by upregulating USP7 utilizing two cell lines, Eca-109 and TE-1 cells [49]. Metformin demonstrated an inhibitory effect on cell proliferation in Eca-109 and TE-1 cells in a dose-dependent manner with a concentration of 5 mM 24 h after treatment. These observations were substantiated by evidence indicating that metformin treatment induced cell cycle arrest. Specifically, both Eca-109 and TE-1 cells exhibited a marked increase in the percentage of cells in the G1/G0 phase and a decreased percentage of cells in the S phase. This was concomitant with alterations in the expression of crucial molecules involved in cell cycle arrest, including the upregulation of p21 and p27 and the repression of cyclin D1 expression. Furthermore, they illustrated the involvement of metformin in the AMPK pathway through the activation of AMP kinase and the subsequent inhibition of mTOR signaling. Notably, the anti-proliferative effects of metformin remained unaffected when the AMPK pathway was inhibited using its antagonist or siRNA oligos. Remarkably, they posited USP7 as a novel molecular protein target of metformin in esophageal cancer cells. This proposition stems from the drug’s specific upregulation of USP7 mRNA and protein levels in both Eca-109 and TE-1 cells, along with its ability to impede cell proliferation by invalidating USP7 through concurrent use of siRNA oligos. USP7 plays crucial roles in the p53 tumor suppressor pathway by stabilizing the p53 protein, thereby exerting control over the expression of various cell cycle regulators, including p21 and p27. Particularly in ESCC, Kobayashi et al. [50] and Damelin et al. [51] disclosed that cells manifest diminished cell proliferation and cell cycle arrest upon exposure to metformin. Their investigation extended to three distinct ESCC cell lines, (T.T, KYSE30, KYSE70) and (WHCO1, WHCO5, and SNO) respectively. Following a 24-h treatment with either 5mM or 10 mM metformin, all cell lines exhibited a notable reduction in cell proliferation and an augmentation in the number of cells in the G0/G1 phase of the cell cycle compared to untreated controls [50,51]. Of paramount significance, Cai et al. while affirming the previously mentioned outcomes regarding ESCC cell proliferation and cycle arrest, additionally demonstrated that the in vivo anti-tumor effects of metformin in an ESCC xenograft model are associated with the upregulation of AMPK, p53, p21CIP1, and p27KIP1, along with the downregulation of cyclin D1 [52]. In the context of EAC, Fujihara et al. demonstrated that metformin hampers EAC cell proliferation by impeding cell cycle progression and altering key molecular pathways by assessing four cell lines, namely OE19, OE33, SK-GT4 and OACM 5.1C [53]. Their study involved a 72-h treatment with 10 mmol/l metformin, providing evidence of suppressed cell proliferation and the blockade of the G0 to G1 transition in the cell cycle. Concomitant with these effects were substantial reductions in G1 cyclins, particularly cyclin D1, cyclin-dependent kinase (Cdk)4, and Cdk6, as well as diminished phosphorylation of retinoblastoma protein. Furthermore, metformin induced notable reductions in the phosphorylation of epidermal growth factor receptor, insulin-like growth factor, and insulin-like growth factor-1 receptor, as well as various angiogenesis-related proteins including vascular endothelial growth factor, tissue inhibitor of metalloproteinase (TIMP)-1, and TIMP-2. Moreover, microRNA expression was markedly altered, with upregulation of three miRNAs and significant downregulation of ten miRNAs out of the 985 miRNA probes investigated [53]. The above are mentioned briefly in the Table 4.

### 3.2. Deciphering the Metformin’s Influence on Apoptosis in EC

Numerous studies have delved into the in vitro and in vivo consequences of metformin on ESCC and EAC cells, with predominant emphasis on two pathways: the signal transducer and activator of transcription 3 (Stat3)/Bcl-2 pathway [54,55,56] and the phosphoinositide 3-kinase (PI3K)/AKT/mTOR pathway [57]. Additionally, more recent studies have explored the anti-tumor activity of metformin through the modulation of redox homeostasis [58,59,60]. In relation to the former mechanism, studies have revealed that metformin demonstrates a selective inhibition of cell growth in ESCC tumor cells while sparing immortalized noncancerous esophageal epithelial cells. Alongside apoptosis, metformin initiated autophagy. The pharmacological or genetic inhibition of autophagy sensitized ESCC cells to metformin-induced apoptotic cell death. Mechanistically, metformin treatment led to the inactivation of Stat3 and its downstream target Bcl-2 [54,55,56]. Consequently, small interfering RNA (siRNA)-mediated Stat3 knockdown heightened metformin-induced autophagy and apoptosis, concurrently intensifying the inhibitory impact of metformin on cell viability. Similarly, metformin repressed the Bcl-2 proto-oncogene, an inhibitor of both apoptosis and autophagy. The ectopic expression of Bcl-2 shielded cells from metformin-mediated autophagy and apoptosis. In Vivo, metformin downregulated Stat3 activity and Bcl-2 expression, induced apoptosis and autophagy, and hindered tumor growth. In summary, the deactivation of the Stat3-Bcl-2 pathway contributes to metformin-induced growth inhibition of ESCC by facilitating crosstalk between apoptosis and autophagy [54,55]. To further corroborate the aforementioned findings, Shaffee et al., by concentering primarily on ESCC TE8 and TE11 cells, substantiated the conclusion that metformin induces apoptosis in ESCC by suppressing Bcl-2 expression and elevating p53 levels [56]. In the context of the PI3K/AKT/mTOR pathway, the anti-hyperglycemic drug has been demonstrated to impede cell cycle arrest in the G0/G1 phase, resulting in apoptosis, activation of caspase 3, downregulation of caspase 9, and an increase in the pro-apoptotic protein Bim [57]. Subsequent investigations revealed that metformin could suppress the expression of insulin-like growth factor 1 receptor and its downstream proteins, including PI3K, protein kinase B (AKT/PKB), phosphorylation of AKT (pAKT), mammalian target of rapamycin (mTOR), p70S6K, and PKM2. Insulin-like growth factor 1 partially reversed metformin-induced apoptosis and mitigated the suppressive effect of metformin on PI3K, pAKT, and PKM2. The knockout of PKM2 resulted in the activation of caspase 3, downregulation of caspase 9, and increased expression of Bim. In an Eca109 xenograft model, metformin significantly curtailed tumor growth. Furthermore, it was observed that metformin treatment increased the rate of apoptosis, downregulated PKM2, and upregulated Bim in tumor tissues [57].

Moreover, in a study conducted by Wang et al., it was revealed that metformin not only demonstrated an anti-proliferative effect in a dose- and time-dependent manner but also had a dose-dependent pro-apoptotic impact on the KYSE450 ESCC cell line [58]. In vivo experiments demonstrated a significant inhibition of KYSE450 xenograft tumor growth with metformin treatment compared to those treated with normal saline, and notably, no discernible toxic reactions were observed. To delve deeper into the underlying mechanism, they found that metformin treatment markedly suppressed the expression of 4EBP1 and S6K1-two target genes of mTOR signaling pathway-in KYSE450 cells both in vitro and in vivo. Additionally, the expression of phosphorylated forms, p-4EBP1, and p-S6K1 in KYSE450 cells, was significantly inhibited both in vitro and in vivo [58].

Recent research has revealed that the anti-diabetic medication metformin demonstrates anti-tumor activity through the modulation of redox homeostasis [55,59,60]. Peng et al. conducted a comparative analysis of the molecular mechanisms of metformin in the ESCC cell line, EC109, and the normal esophageal epithelial cell line, HEEC [55]. Metformin exhibited more pronounced inhibitory effects on cell proliferation in EC109 cells compared to HEECs. The drug induced apoptosis in EC109 cells in a dose-dependent manner, contrasting with the response in HEECs. The expression of Stat3, both at the mRNA and protein levels, was higher in EC109 cells than in HEECs. Subsequent investigation indicated that metformin could mitigate the phosphorylation of Stat3 and the expression of Bcl-2, a process partially restored by IL-6 in EC109 cells but not in HEECs. Conversely, metformin increased the level of reactive oxygen species (ROS) in both cell lines, yet this intracellular ROS variation did not impact apoptosis. Notably, metformin exhibited distinct functional roles in apoptosis between esophageal carcinoma cells and normal esophageal cells. Consequently, the Stat3/Bcl-2 pathway-mediated apoptosis underscores the cell type-specific sensitivity to the drug, suggesting that metformin possesses therapeutic efficacy and selectivity in the context of esophageal cancer [55]. Concentrating on EAC cells, Hong et al. conducted an investigation into the mechanistic role of the tyrosine kinase receptor AXL in autophagy, as well as the effects induced by metformin on the growth and survival of EAC [60]. They revealed that AXL orchestrates autophagic flux through the activation of AMPK-ULK1 signaling in an ROS-dependent manner induced by glucose starvation. AXL positively modulates basal cellular ROS levels without significantly influencing mitochondrial ROS production in EAC cells. Furthermore, they demonstrated that AXL expression is essential for metformin-induced apoptosis in EAC cells in vitro. The induction of apoptosis by metformin is notably diminished by the inhibition of autophagy through the genetic silencing of Beclin1 or ATG7 autophagy mediators, thereby confirming the necessity of intact autophagy for enhancing metformin-induced apoptosis in EAC cells. The data from their study indicate that metformin-induced autophagy serves a pro-apoptotic function in EAC cells. Additionally, they substantiated the conclusion that the metformin-induced suppression of tumor growth in vivo is highly contingent on AXL expression, as demonstrated in a tumor xenograft mouse model of EAC [60]. The above are presented concisely in Table 5.

### 3.3. Metformin’s Effects on Angiogenesis, Invasion, and Metastasis

Tumor metastasis is associated with intricate interactions among primary tumor cells, encompassing activities such as invasion, intravasation, immune evasion, and extravasation from the circulatory system. Subsequent events include lymphangiogenesis/angiogenesis and migration towards specific target organs [61].

Metastasis-associated colon cancer-1 (MACC1), a gene associated with tumor metastasis, plays a pivotal role in the development of cancer, although its specific functions and mechanisms in ESCC remain elusive. Recent investigations have yielded insights into the overexpression of MACC1 in ESCC and its correlation with lymph node metastasis. Consequently, the suppression of MACC1 through knockdown has been found to inhibit cell proliferation and metastasis while promoting apoptosis in ESCC Eca9706 and Kyse150 cells [62]. Furthermore, MACC1 knockdown impedes ESCC cell autophagy, and concurrent treatment with 3-methyladenine, an autophagy inhibitor, mitigates the MACC1-induced malignant phenotype in ESCC cells. Additionally, MACC1 knockdown deactivates the AMPK-ULK1 signaling pathway, and the use of metformin, an AMPK activator, rescues MACC1-induced autophagy in ESCC cells [62].

Angiogenesis is a vital process for tumor growth and metastasis, exerting significant control over cancer progression. The tumor vasculature consists of an irregular and disorganized network of vessels, distinct from the organized and functional vessels found in normal tissue. Research has identified marked differences at the molecular and functional levels between tumor endothelial cells (TECs) and normal endothelial cells (NECs). By comparing gene expression profiles, specific markers exclusive to TECs have been discovered. Despite the recognized influence of the tumor microenvironment on angiogenesis, the precise underlying mechanisms remain unclear [63]. To investigate the impact of the microenvironment in human ESCC, Yang et al. used supernatants from KYSE450 or KYSE70 cultures and ESCC tissue homogenates, collectively known as tumor-conditioned medium (TCM). Their objective was to understand the mechanism behind TCM-induced angiogenesis and its influence on NECs. The findings revealed that TCM stimulated enhanced angiogenic properties in NECs, including migration, invasion, and tube formation. Notably, the TCM-induced NECs expressed higher levels of TEC markers, indicating the potential of TCM to drive NECs towards a TEC-like state [63]. Furthermore, through microarray gene expression analysis, significant genomic alterations were observed in NECs exposed to TCM. These alterations affected numerous regulatory networks, with a particular impact on the c-MYC and JAK/STAT3 signaling pathways. Importantly, they explored the anti-angiogenic effects of metformin and discovered its ability to impede the transition of NECs towards TECs induced by the ESCC microenvironment. Metformin achieved this by inhibiting the JAK/STAT3/c-MYC signaling pathway. In a pioneering demonstration, the anti-proliferative and anti-angiogenic activity of metformin was further validated in a human ESCC patient-derived xenograft (PDX) mouse model [63]. Furthermore, over the past decade, extensive research has been conducted to investigate the effectiveness of metformin in impeding the migration and invasion of ESCC cells [64]. In particular, metformin has been found to hinder the migration and invasion of ESCC EC109 cells, as well as to suppress nuclear factor-κB activation, MMP-9 expression, and N-cadherin expression in a manner dependent on phosphorylated AKT. These findings suggest that metformin exerts its inhibitory effects on the migration and invasion of human ESCC cells by attenuating AKT phosphorylation and modulating the expression of genes associated with migration and invasion [64]. Moreover, an additional investigation [65] illustrated the dose-dependent and time-dependent anti-invasive and anti-metastatic properties of metformin on human ESCC cell lines EC9706 and Eca109. These effects were observed both in vitro and in vivo, likely attributable to the downregulation of MMP-2 and MMP-9. These enzymes serve as pivotal regulators in the context of cancer cell invasion and metastasis, as highlighted in the aforementioned study [65].

Fan et al. employed in vitro and in vivo experiments to further expound on the influence of metformin on the tumor microenvironment through the activation of the AMPK pathway [66]. They delved into the effects of metformin on the carcinogenesis of ESCC using a rat ESCC model. The outcomes revealed a significant reduction in the incidence and precancerous lesions of ESCC, along with the inhibition of proliferation and promotion of apoptosis in esophageal epithelial cells in rats treated with N-nitroso-N-methylbenzylamine (NMBzA). Additionally, metformin exhibited anti-cancer effects by increasing apoptosis and suppressing migration, colony formation, and tumor sphere formation in human ESCC cells in vitro. Metformin treatment was associated with the activation of AMPK, which, in turn, attenuated the signaling of downstream molecules such as p-mTOR, p-p70S6K, and cyclin D1 expression, both in vivo and in vitro. These findings underscore the chemopreventive potential of metformin in the context of ESCC carcinogenesis [66].

### 3.4. Metformin-Mediated Epigenetic Changes: Understanding the Molecular Mechanisms

The collective findings of several studies shed light on the multifaceted roles of metformin and various molecular pathways in ESCC, providing a foundation for potential therapeutic strategies and biomarker development in the management of this challenging cancer. Conversely, a singular investigation, as denoted by a single study (reference omitted for brevity), explored the impact of metformin on the epigenetics of EAC.

Kobayashi et al. assessed alterations in microRNA (miRNA) expression profiles within the human ESCC cell lines T.T, KYSE30, and KYSE70 [50]. In KYSE30 cells treated with 10 mM metformin in vitro, a custom microarray analysis of 985 human miRNA probes identified significant changes in miRNA expression. After 72 h, 17 miRNAs were upregulated, and 45 miRNAs were downregulated compared to untreated cells. Unsupervised clustering analysis illustrated a distinct miRNA expression profile in metformin-treated KYSE30 cells [50]. Additional studies examined the gene expression profiles of numerous miRNAs in ESCC cell lines [67,68,69]. These analyses employed microarray data in conjunction with the Connectivity Map (CMAP) database, revealing crucial biological functions associated with these miRNAs. These functions encompassed processes such as development, differentiation, apoptosis, and proliferation, elucidating the intricate connections between metformin, genes, and the pathogenesis of ESCC [67,68,69] In particular, metformin might activate the TMCO3 and PLA2G4A genes, which are tumor suppressor genes upregulated by miR-375 and have a tumor-suppressive function in themselves as an alternative substance to miR-375 [67]. Investigating the communication between hypoxic and normoxic cells via exosomes, this study identified exo-miR-340-5p as a key player in the transfer of radioresistance. Metformin was found to enhance radiosensitivity by targeting the miR-340-5p/KLF10 axis, presenting a potential strategy for overcoming radiotherapy resistance [68]. Additionally, Wang et al. investigated the evasion of apoptosis as a major contributor to chemo- and radiotherapy resistance in ESCC [69]. The study demonstrated that metformin induces pyroptosis, a non-apoptotic programmed cell death, in ESCC, particularly in its advanced stages. The scaffolding oncogene proline-, glutamic acid-, and leucine-rich protein-1 (PELP1) was found to be upregulated in advanced ESCC stages and associated with cancer progression. Metformin-induced gasdermin D (GSDMD)-mediated pyroptosis was observed, and its effects were counteracted by forced expression of PELP1. The mechanism involves metformin targeting the miR-497/PELP1 axis. In conclusion, metformin and other pyroptosis-inducing agents are potential alternatives for treating chemo- and radiotherapy-resistant ESCC and other cancers sharing similar pyroptosis mechanisms [69]. The evidence outlined above is succinctly summarized in Table 6.

### 3.5. Enhancing Chemotherapeutic Efficacy: Synergies of Metformin in Combination Therapies

Contemporary therapeutic approaches employed in EC demonstrate limited efficacy and frequently manifest both chemotoxicity and chemoresistance. Explorations into more efficacious targeted treatment alternatives through monoclonal antibody therapies exhibit considerable promise [71,72]. Nevertheless, accessibility to such therapies in developing countries remains exceptionally restricted, chiefly due to financial constraints. Consequently, an ongoing and imperative need persists for alternative, effective, and cost-effective treatment options.

#### 3.5.1. Metformin and Cisplatin

Although metformin commonly diminishes cell proliferation across various cancer types, it seldom induces apoptosis. Consequently, it is being investigated in conjunction with conventional chemotherapeutic agents such as cisplatin. The outcomes of this combined treatment vary, as certain studies indicate that metformin may augment the efficacy of chemotherapeutic drugs, while others demonstrate an elevated chemoresistance in the presence of metformin [73,74]. Cisplatin exerts cytotoxicity through various mechanisms, encompassing the formation of DNA and protein adducts, as well as induction of oxidative stress. Numerous resistance mechanisms to cisplatin have been identified, with one notable example being the sequestration of cisplatin by glutathione, a significant component of intracellular thiols [75].

Of considerable significance, metformin assumes a protective role against cisplatin cytotoxicity in ESCC cells [51]. Significantly, metformin protects against cisplatin cytotoxicity in ESCC cells, not solely by reducing cell proliferation. It offers minimal to partial protection against mitomycin C, relying on enhanced glycolysis, increased NAD(P)H levels, and elevated intracellular thiols. This coincides with reduced cisplatin–DNA adduct formation. Inhibition of glutathione synthesis reverses this protection, emphasizing glutathione’s role in cisplatin detoxification by metformin-treated cells. Compounds like copper-bis (thiosemicarbazones), copper diacetyl-bis(4-methylthiosemicarbazonato) copper (II) (Cu-ATSM), and copper glyoxal-bis(4-methylthiosemicarbazonato)copper(II) (Cu-GTSM), sequestered under reducing conditions, show cytotoxicity. Combining metformin with these drugs has emerged as an innovative therapeutic strategy for ESCC treatment [51].

The efficacy of the metformin–cisplatin combination was further substantiated by Wang et al., who conducted a study concentrating on the treatment of ESCC KYSE450 cells [58]. The study involved the co-administration of metformin (at concentrations of 0, 5, 10, and 20 mmol/L) and cisplatin (at 2 mg/L) over 24, 48, and 72 h. The findings demonstrated that metformin augmented the antiproliferative impact of cisplatin, resulting in a noteworthy increase in the percentage of apoptotic cells, in a dose- and time-dependent manner [58]. Conversely, Ju et al. conducted the first study elucidating that metformin synergistically enhances cisplatin cytotoxicity in the ESCC ECA109 cell line under conditions of glucose deprivation [76]. This condition is deemed to more accurately simulate the microenvironment within solid tumors. Notably, this observed effect markedly differs from the previously documented cytoprotective impact of metformin against cisplatin when applied in the commonly used high-glucose incubation medium. The potential mechanisms contributing to the synergistic effect of metformin on cisplatin-induced cytotoxicity under glucose deprivation conditions may encompass the amplification of metformin-associated cytotoxicity, substantial reduction in cellular ATP levels, deregulation of the AKT and AMPK signaling pathways, and impairment of DNA repair functionality [76].

#### 3.5.2. Metformin and 5-Fluorouracil

Honjo et al. evaluated the impact of metformin, either alone or in conjunction with 5-fluorouracil (5-FU), on the survival and apoptosis of multiple EC cell lines, encompassing both EAC lines (FLO-1, BE3, SKGT-4, OE33, JHESO, OACP) and ESCC lines (YES-6, KATO-TN) [77]. Metformin reduced cell survival and induced apoptosis in EC cell lines. Combining metformin with 5-FU increased EC cell sensitivity to 5-FU’s cytotoxic effects. Metformin decreased the expression of oncogenes, including those in the PI3K/mTOR pathway, as well as survival and cancer stem cells. Immunoblots and transcriptional analyses confirmed dose-dependent downregulation. Metformin preferentially reduced tumor sphere formation by ALDH-1+ cells. In vivo experiments showed metformin effectively reduced tumor growth, and in combination with 5-FU, a synergistic reduction occurred. The study suggests metformin inhibits EC cell growth and enhances sensitivity to 5-FU by targeting cancer stem cells and mTOR pathway components, supporting its potential benefits for EC patients in combination therapies [77].

#### 3.5.3. Metformin and Ionizing Radiation

While concurrent chemoradiotherapy significantly improves the treatment outcomes for unresectable EC, it is associated with notable adverse reactions. Several studies [78,79] have delved into the synergistic interplay between metformin and ionizing radiation (IR), revealing that metformin effectively enhances the anti-proliferative effects of IR on EC cells, particularly ECa109 cells. The combination resulted in more pronounced DNA damage, evident through the detection of γH2AX foci, a well-established marker of IR-induced DNA double-strand breaks (DSBs). Remarkably, these foci exhibited a delayed resolution within 24 h, indicating a slowdown in DNA repair when metformin was combined with radiation [78]. Moreover, the combination of metformin and radiation synergistically induced apoptosis and cell cycle arrest in ECa109 cells. The potential mechanisms underpinning metformin’s sensitization of ECa109 cells to IR involve the targeting of the ATM and AMPK/mTOR/HIF-1a pathways. This suggests a promising avenue for enhancing the efficacy of radiation therapy in EC treatment while potentially mitigating adverse reactions associated with conventional chemoradiotherapy [78]. Additionally, the inhibitory effect of metformin on IR-induced epithelial-to-mesenchymal transition (EMT) via the suppression of the transforming growth factor-β (TGF-β)-Smad phosphorylation pathway, as well as a component of the non-Smad pathway, has been examined. Biguanide metformin is reported to prevent transforming growth factor-β (TGF-β)-induced EMT and proliferation of cancer. Nakayama et al. mentioned that after IR exposure, TE-9 ESCC cells presented an altered morphology and lost cell–cell adhesion [79]. Pre-treatment with 0.5 mM metformin inhibited these morphological changes, and, when administered without IR, metformin did not induce any morphological changes when compared with the control. Moreover, it is well observed that IR induces expression of mesenchymal markers (vimentin and N-cadherin), EMT- associated transcription factors (Slug, Snail, and Twist), and matrix metalloproteinases, enhances the invasive potential and migratory capacity of TE-9 cells and increases the expression of hypoxia-related factor-1α (HIF)-1α and TGF-β1. The introduction of metformin mitigates the effects of IR on mesenchymal marker protein expression and suppresses IR-induced phosphorylation of Smad2 and Smad, although without affecting the expression of HIF-1a and TGF-β1. Notably, the combined treatment of IR and metformin results in the enhanced phosphorylation of AMP-activated protein kinase, but metformin alone counteracts the IR-induced phosphorylation of mammalian target of rapamycin [79].

#### 3.5.4. Metformin and Disulfiram

More recently, disulfiram (DSF) and copper-DSF (Cu-DSF) have been discovered to demonstrate potent anti-cancer activity through various mechanisms. These include inhibiting the activating transcription factor/cyclic AMP-responsive element binding protein (ATF/CREB), nuclear factor kappa-light-chain-enhancer of activated B cells (NF-κB), P-glycoprotein, DNA topoisomerases, and DNA methyltransferases, as well as acting as a robust inhibitor of the proteasome, invasion, and angiogenesis [80]. In a study by Jivan et al., Cu-DSF and DSF exhibited significant cytotoxicity in a panel of ESCC cells (WHCO1, WHCO5, SNO), and the addition of metformin significantly enhanced the effects of DSF. It was observed that elevated copper transport contributed to the cytotoxicity induced by DSF and metformin-DSF, as the use of a cell-impermeable copper chelator, bathocuproinedisulfonic acid, partially reversed the cytotoxic effects of these drugs. Interestingly, metformin-treated ESCC cells exhibited higher intracellular copper levels. Moreover, the high dependence of cancer cells on protein degradation/turnover pathways might make them preferential targets for DSF, and metformin further enhances DSF’s role as a proteasome inhibitor. Additionally, this acid-labile compound reduces lysosomal acidification, and co-treatment of DSF and metformin interferes with the progression of autophagy in these cells, suggesting that the lysosome serves as a target for DSF [80].

#### 3.5.5. Metformin and CB-839 (Glutaminase 1 Inhibitor)

Cellular metabolism reprogramming is considered a key aspect of tumorigenesis, with two main forms of dysregulated metabolism observed: the Warburg effect and active glutaminolysis [81]. Within tumor cells, glutamine (Gln) undergoes dynamic metabolic processes, contributing significantly to diverse functions. These include: (1) the generation of energy through α-KG within the tricarboxylic acid (TCA) cycle; (2) involvement in biosynthetic pathways, including de novo purine and pyrimidine synthesis, along with the production of non-essential amino acids; and (3) a role in the synthesis of reductive equivalents, such as glutathione [82].

The study focuses on the role of the Fbxo4-cyclin D1 axis, commonly dysregulated in various cancers, in controlling glutamine addiction (Gln addiction) independently of known signaling pathways [83]. Tumor cells with dysregulated Fbxo4-cyclin D1 show increased Gln uptake, leading to a paradox of elevated Gln consumption but compromised energy production due to mitochondrial dysfunction [83]. This intrinsic vulnerability provides a therapeutic opportunity. Cyclin D1, frequently overexpressed in cancers, is regarded as an oncogenic driver. CDK4/6 inhibitors, such as palbociclib, show initial efficacy, but resistance develops, often associated with Rb loss [84].

Recent investigations have shown that Gln metabolism plays a crucial role in the survival and proliferation of tumor cells, making them dependent on Gln [85]. Glutaminolysis, the process by which Gln is metabolized to produce energy, is regulated by oncogenes and tumor suppressors through the control of glutaminase (GLS) expression and activation. GLS1, one of the isoforms of GLS, is a key enzyme in this process. Suppression of GLS1 leads to apoptosis, decreased cell proliferation, and inhibited tumor growth. Additionally, dysregulation of the Fbxo4-cyclin D1 axis, caused by Fbxo4 loss or cyclin D1 amplification, results in Gln addiction and mitochondrial dysfunction. In a clinical setting, combined treatment with a GLS1 inhibitor (CB-839) and metformin/phenformin shows promise in inducing apoptosis and suppressing cell proliferation, particularly in tumors resistant to CDK4/6 inhibitors. Metformin, known for compromising mitochondrial oxidative phosphorylation (OXPHOS), can further reduce energy production in Gln-depleted or GLS1-suppressed cancer cells, activate AMPK, suppress mTORC1, and reprogram metabolism towards glutaminolysis. Based on these findings, targeting glutaminolysis and OXPHOS is suggested as an effective therapeutic approach for ESCC with a dysregulated Fbxo4-cyclin D1 axis and for overcoming palbociclib resistance. The study by Qie et al. emphasizes the importance of disrupting both glutaminolysis and mitochondrial respiration to treat ESCC and overcome palbociclib resistance [85]. The above are illustrated in Figure 2.

#### 3.5.6. Metformin and 2-Deoxy-d-glucose (2DG)

Mitochondrial dysfunction and aerobic glycolysis are characteristic features of aggressive cancer [56]. These alterations pose significant challenges for cancer treatment, causing resistance to chemotherapy and radiation in low-oxygen environments. Cancer cells heavily rely on glycolysis for energy production, presenting an opportunity to preferentially kill them by inhibiting glycolysis. 2-DG and metformin are two agents that disrupt cell metabolism and signaling pathways, depleting ATP, inducing autophagy and impacting cell survival. The expansion of ESCC is associated with genetic and epigenetic changes, including activation of oncogenes, inactivation of tumor suppressor genes, and mutations in p53. The altered expression of Bcl-2 family proteins is also implicated in oncogenesis and primitive lesions. In the case of ESCC, metformin and 2DG, either alone or in combination, induce apoptosis in cell lines by activating p53 and downregulating Bcl-2 expression [56].

Investigating the role of cancer stem cells (CSCs) in ESCC progression, esophageal CSCs exhibit distinct metabolic features (higher glycolysis and oxidative phosphorylation) regulated by the Hsp27–AKT–HK2 pathway [70]. Inhibition of CSC metabolism, through treatment with 2-DG and metformin, showed promise in reducing cell growth and tumor formation, providing a potential target for therapeutic intervention [70].

#### 3.5.7. 3-Aminobenzamide (3-ABA) Combined with Cisplatin or Metformin

Elevated SOX2 levels in ESCC are closely linked to increased incidence [86]. SOX2, crucial for maintaining squamous cell identity, frequently exhibits mutations in ESCC, contributing to tumor formation and drug resistance. Poly (ADP-Ribose) polymerase 1 (PARP1), known for its role in DNA repair, interacts with SOX2 and influences stem cell functions, including pluripotency and redox homeostasis. Inhibiting PARP1 reduces ESCC proliferation and impacts various cancer-related signaling pathways. Combining the PARP1 inhibitor 3-aminobenzamide (3-ABA) with cisplatin synergistically suppresses ESCC cell growth. Notably, this suppressive effect is potentiated by metformin. Targeting PARP1, a binding partner of SOX2, emerges as a promising therapeutic strategy for individuals with elevated SOX2 levels, offering potential avenues for ESCC treatment [86]. The above are summarized in Table 7.

## 4. Understanding Metformin’s Impact on Immune Responses in EC

Beyond its well-documented impact on proliferation and apoptosis, metformin showcases additional dimensions of efficacy in EC. Emerging evidence suggests that metformin’s immunomodulatory properties play a crucial role in influencing the EC TME, contributing to a more comprehensive approach in addressing EC progression. Understanding these diverse facets positions metformin as a promising candidate for integrated therapeutic strategies against EC.

### 4.1. Navigating Immune TME: Metformin’s Role in Esophageal Cancer

In recent studies, compelling evidence has emerged, shedding light on the intricate relationship between the immune TME and metformin in the context of EC. Takei et al. explored the development of EC in a rat model with chronic GERD triggered by exposure to bile acid and chronic inflammation [87]. Takei et al. examined immune cell dynamics during different phases of esophageal carcinogenesis and their response to metformin treatment [87]. While the infiltration of CD3^+^ T cells peaked at 20 wps, metformin notably increased CD3^+^ T cell numbers at 10 and 40 wps. Metformin treatment during the inflammatory phase (10 wps) significantly elevated CD4^+^ and CD8^+^ T cell numbers. The transition from inflammation to carcinogenesis in non-treated controls saw a rise in Treg cell percentages, which metformin counteracted by decreasing Treg and Th17 cell percentages at 20 and 40 wps. They also explored alterations in macrophage populations. In the non-treated group, CD11b^+^CD68^+^CD86^+^ M1s increased at 20 wps and decreased at 40 wps, while metformin treatment consistently upregulated M1s from 10 to 40 wps. Metformin effectively shifted the M1s/M2s balance in favor of M1s, particularly at 20 wps. CD11b^+^CD68^+^CD163^+^ M2s increased at 40 wps in non-treated controls but significantly decreased with metformin treatment at all stages. Interestingly, metformin downregulated TNF-α expression in M1s at 20 wps but paradoxically upregulated it at 40 wps. IFN-γ levels in M1s were slightly decreased at 40 wps with metformin. The nuanced effects of metformin extended to macrophage cytokine expression. TNF-α expression in M1s reflected the percentage of M1s, and similar patterns were observed for IFN-γ, TGF-β, IL-10, and M2 percentages. Metformin inhibited the upregulation of macrophage p-Stat3 (Ser727) levels at 40 wps [87]. These findings collectively underscored the intricate immunomodulatory impact of metformin on T cell and macrophage populations, shedding light on potential mechanisms influencing the esophageal carcinogenic process.

Quin et al. explored the mechanisms by which metformin inhibits the accumulation of MDSCs in the TME [88]. Metformin reduces MDSC migration in patients with ESCC. The key findings indicated that the frequency of tumor-infiltrated polymorphonuclear (PMN)-MDSCs was elevated, correlating with poorer prognosis in ESCC. PMN-MDSCs demonstrated immunosuppressive activity in vitro. Metformin treatment effectively reduced MDSC migration in patients. Mechanistically, metformin inhibited CXCL1 secretion in ESCC cells and tumor xenografts. This effect was mediated by enhanced AMPK phosphorylation and induced DACH1 expression, leading to NF-kB inhibition and consequent reduction in MDSC migration. They also demonstrated that knockdown of AMPK and DACH1 expression counteracted the inhibitory effect of metformin on MDSC chemotaxis [88]. These findings highlight a novel anti-tumor effect of metformin mediated by the AMPK/DACH1/CXCL1 axis, offering potential therapeutic implications in cancer treatment.

The compelling evidence from preclinical studies, showcasing the immune microenvironment reprogramming by low-dose metformin in ESCC, paved the way for subsequent human clinical trials [32]. In a phase II clinical trial for ESCC, low-dose metformin (250 mg/day) demonstrated no direct impact on tumor cell proliferation but effectively reprogrammed the ESCC immune TME, shifting it towards an “infiltrated–inflamed” state. This was characterized by increased CD8^+^ cytotoxic T lymphocytes, CD20^+^ B lymphocytes, and tumor-suppressive (CD11c+) macrophages, along with a decrease in tumor-promoting (CD163^+^) macrophages. Metformin also enhanced macrophage-mediated phagocytosis of ESCC cells in vitro. Metformin also triggered AMPK activation and STAT3 inactivation, influencing effector cytokine production. This suggests that low-dose metformin has the potential to reprogram the tumor immune microenvironment, making it a potential candidate for immune response modulation in the treatment of ESCC [32]. The above are illustrated in brief in Figure 3.

Collectively, preclinical evidence revealing immune microenvironment reprogramming by metformin in EC [87,88] led to human clinical trials [32]. Results demonstrated metformin’s capacity to reshape the tumor immune TME in ESCC patients, emphasizing its potential in clinical applications.

### 4.2. The Effects of Metformin in Inflammatory Signaling

Accumulating evidence suggests a potential role for metformin in modulating inflammation-related signaling pathways in EC. Studies have indicated that metformin may exert anti-inflammatory effects by inhibiting the NF-κB pathway [64,89] activating AMPK [66], and influencing the secretion of anti-inflammatory cytokines. Additionally, research has explored metformin’s impact on non-apoptotic programmed cell death (PCD) induction with consequent effects in the EC TME [69].

As mentioned above, Wang et al. investigated the potential of metformin in inducing pyroptosis, a form of PCD, in ESCC [69]. They reported that metformin treatment triggered pyroptosis both in vitro and in vivo. The scaffolding oncogene PELP1, associated with cancer progression, was found to be upregulated in advanced ESCC stages. Intriguingly, metformin-induced pyroptosis involved GSDMD and was mitigated by the forced expression of PELP1. The mechanism underlying this effect was associated with the miR-497/PELP1 axis [69]. The findings suggest that metformin, by inducing pyroptosis, could serve as an alternative treatment for chemo- and radiotherapy refractory ESCC. As regards NF-κB signaling, metformin’s inhibition of NF-κB activation has been observed in various studies in EC [64,89]. According to Sekino et al. the change in the intracellular localization of NF-κB by metformin may indicate that metformin exerts its antitumor effect by inhibiting EMT through NF-κB, thereby downregulating the progression of ESCC [89], while He et al., taking a step further, reported that metformin was found to inhibit migration and invasion of esophageal cancer cells (EC109) by downregulating the expression of phosphorylated AKT (p-AKT) and NF-κB (p65) [64]. Moreover, metformin upregulated the mRNA expression of numerous genes, including heat shock protein family A (Hsp70) member 6 (HSPA6), a cancer immune-related gene [90]. HSPA6 expression correlates with disease-free survival (DFS) of the patients with all stage ESCC, especially with stage I/II ESCC. Low HSPA6 expression is an independent poor prognostic factor of stage I/II ESCC. Therefore, HSPA6 could be used as a potential biomarker for the recurrence risk of stage I/II ESCC [90]. Wang et al. underscored metformin’s pivotal role in inhibiting nicotine-enhanced cancer growth in ESCC [91]. Metformin disrupted the cholinergic receptor nicotinic alpha 7 subunit (CHRNA7), a key player in nicotine-induced oncogenesis. By downregulating CHRNA7 expression and counteracting nicotine-induced DNA hypomethylation, metformin effectively inhibited cancer-initiating cell properties. Notably, it targeted the JAK2/STAT3/SOX2 signaling pathway, which is frequently dysregulated in human ESCC and is activated by CHRNA7. This dual action on CHRNA7 and the JAK2/STAT3/SOX2 pathway positions metformin as a promising strategy for combating ESCC progression influenced by nicotine [91]. Finally, metformin downregulated the expression of pro-inflammatory cytokines. Fan et al. demonstrated on an ESCC rat model induced with N-nitroso-N-methylbenzylamine (NMBzA) that metformin reduced esophageal inflammation and carcinogenesis by suppressing iNOS, COX-2 and IL-6 expressions. The latter was mediated by upregulation of AMPK, which downregulated mTOR activity [66]. Lu et al. took a step further demonstrating that metformin inhibited PD-L1 expression in ESCC by blocking the IL-6/JAK2/STAT3 signaling pathway. In ESCC cell lines, metformin significantly inhibited PD-L1 expression through the IL-6/JAK2/STAT3 pathway, not the canonical AMPK pathway. In co-culture systems, metformin enhanced T cell activation and killing function. In vivo experiments confirmed metformin’s downregulation of PD-L1 and combined treatment with metformin and PD-1 inhibitors synergistically enhanced the antitumor immune response [92]. Overall, metformin’s impact on ESCC involves suppressing PD-L1 via the IL-6/JAK2/STAT3 pathway, improving the antitumor immune response. The above are briefly illustrated in Figure 4.

In summary, metformin exerts various anti-inflammatory effects in esophageal cancer, underscoring its potential as a therapeutic agent in addressing ESCC.

## 5. Discussion

Chemotherapy strategies for esophageal cancer were initially developed for SCC, but with the rise of adenocarcinomas, treatment approaches have converged. Notably, clinical trials from the mid-1990s onwards have included patients with gastric, esophageal, or EGJ cancer, regardless of histology [71]. While the histologic subtype initially did not significantly impact response rates or survival with cytotoxic chemotherapy, recent advances in understanding genomic alterations are revealing differences between SCC and EAC [93,94]. With the advent of molecularly directed therapy and immunotherapy, treatment trajectories are diverging once again [95,96]. Therapies targeting specific markers like HER2 and VEGF are applicable only to adenocarcinomas, while immunotherapy, particularly immune checkpoint inhibitors, show promise for SCC [97]. For EGA, biomarker assessment, including HER2, PD-L1, and mismatch repair status, guides treatment decisions. Trastuzumab is recommended for HER2-overexpressing EAC, while ICIs show efficacy in HER2-negative adenocarcinomas with high or intermediate PD-L1 expression or deficient mismatch repair [98]. EACs are further stratified based on HER2 expression, PD-L1 status, and mismatch repair status, influencing the choice between chemotherapy alone, chemotherapy plus immunotherapy, or targeted therapies [98]. In the case of ESCC, chemotherapy combined with immunotherapy is preferred, showcasing greater activity compared to chemotherapy alone, particularly in patients with higher PD-L1 expression [72,99].

In a groundbreaking study, Zhao et al. presented a comprehensive characterization of DDR gene expression in a Chinese cohort of ESCC patients with clinical follow-up data [100]. The analysis classified patients into DDR^active^ and DDR^silent^ subtypes, revealing unique molecular features specific to locoregional ESCC. Low BRCA1 and high HFM1 expression emerged as independent prognostic biomarkers for poor survival in locoregional ESCC, shedding light on potential treatment options post-esophagectomy. Pathway-level comparison disclosed distinct molecular signatures between DDR^active^ and DDR^silent^ tumors, suggesting a link between DDR deficiency and immune response modulation. DDR^silent^ tumors exhibited higher immune infiltration, TEX signal gene set enrichment, and PD-1 expression, indicating a potential for immune checkpoint inhibitor therapy. Notably, GITR triggering and BTLA blockade, in combination with PD-1 blockade, are proposed as immunotherapy strategies for DDR^silent^ ESCC. While functional assays support the efficacy of anti-PD-1-based combination immunotherapy, further mechanistic and clinical studies are warranted. The study underscores the significance of DDR subtyping in prognostication and therapeutic stratification for locoregional ESCC, offering insights into potential immunotherapy approaches beyond anti-PD-1 antibodies. These findings lay the groundwork for future therapeutic developments and clinical trials to enhance outcomes in locoregional ESCC patients. Regarding EAC, the Neo-AEGIS study aimed to compare trimodality therapy with perioperative chemotherapy for adenocarcinoma of the esophagus and EGJ [101]. Despite not completing recruitment, the study provided valuable insights. With a median follow-up over 3 years, no significant differences in OS were observed between the two approaches. The study noted fewer pathologically complete responses and lower R0 rates in the perioperative chemotherapy group, but the pattern of failure and postoperative outcomes did not differ significantly [101]. The trial did not establish non-inferiority, and the upper bound of the 95% CI exceeded the predefined limit. However, the data, though inconclusive statistically, present the largest randomized series comparing these approaches. The study suggests continued equipoise in decision-making for this cancer, with survival outcomes consistent with current literature. Ongoing and future trials, especially those exploring immunotherapy combinations, will contribute to the evolving landscape of treatment for locally advanced adenocarcinoma of the esophagus and EGJ [101].

The findings from the Neo-AEGIS study underscore the need to expand our arsenal against esophageal cancer. Metformin has been studied in various cancer contexts [9], including esophageal cancer, for its potential anti-cancer effects. The studies examining metformin’s impact on EC face several limitations that warrant consideration. Firstly, a significant body of the evidence is derived from retrospective and observational studies, introducing challenges associated with biases, confounding variables, and the inability to establish causation. Secondly, the heterogeneity in study populations, including variations in demographics, cancer stage, and comorbidities, complicates the generalization of results across diverse patient groups. Moreover, the limited number of randomized controlled trials focused on metformin and esophageal cancer hinders the establishment of robust causal relationships. Finally, limited diversity in study populations and the absence of well-designed trials with extended follow-up impede the generalizability and depth of current knowledge on metformin’s role in EC. Overall, while there is interest in the potential benefits of metformin, addressing these limitations through research is crucial for a more in depth understanding of its efficacy in EC prevention and treatment.

## 6. Conclusions—Future Directions

As we navigated the complex landscape of EC treatment, further investigations are warranted to delineate optimal dosages, treatment durations and patient stratifications for metformin therapy. Moreover, exploring synergistic effects with existing therapeutic modalities and investigating metformin’s impact on specific molecular pathways could uncover novel avenues for personalized treatment strategies. The integration of metformin into the evolving landscape of EC management holds great potential and future directions should focus on translating these findings into clinical advancements for improved patient outcomes.

## Figures and Tables

**Figure 1 ijms-25-02978-f001:**
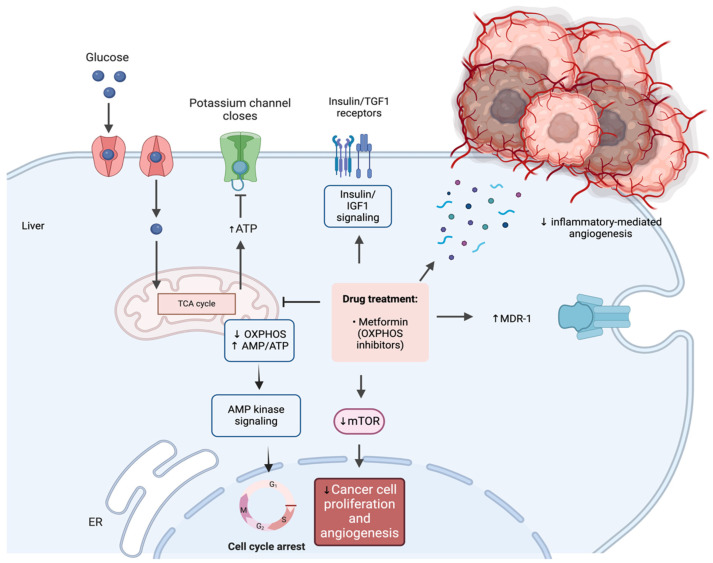
The multifaceted mechanisms underlying metformin’s anti-cancer effects. Metformin, recognized for its role in enhancing insulin sensitivity and reducing hepatic glucose output, engages various pathways. It activates AMPK, inhibits mTOR and lipogenic enzymes, influences insulin/IGF levels, and impacts signaling pathways like Ras/Raf/MAPK, NF-kB, and HER2. ESCC, esophageal squamous cell carcinoma; AMPK, 5′ AMP-activated protein kinase; mTOR, the mammalian target of rapamycin protein; ATP, adenosine triphosphate; OXPHOS, oxidative phosphorylation; MDR-1, multidrug resistance mutation 1; IGF1, insulin-like growth factor. Created with Biorender.com (accessed on 9 February 2024).

**Figure 2 ijms-25-02978-f002:**
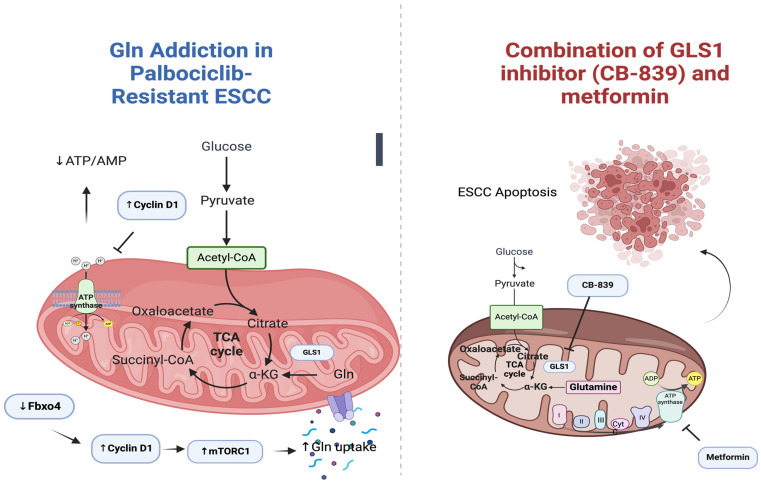
A schematic illustration of palbociclib-resistant ESCC cells which exhibit metabolic reprogramming marked by Gln addiction, rendering them more susceptible to the combined treatment. Created with Biorender.com (accessed on 29 February 2024).

**Figure 3 ijms-25-02978-f003:**
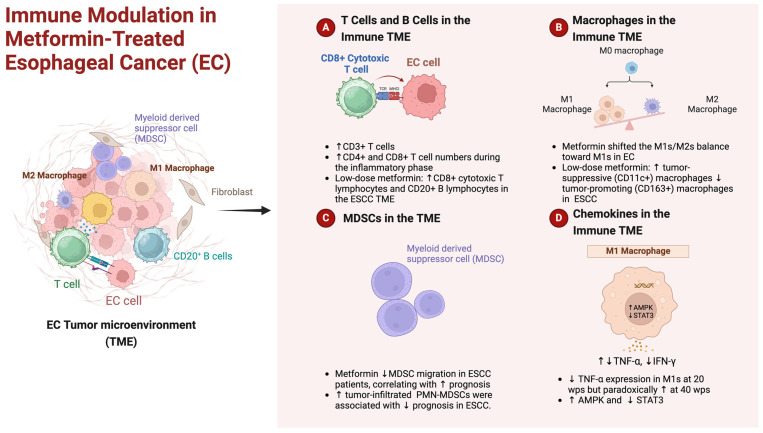
The metformin effects on the EC immune TME. It boosts CD3+ T cells, shifts macrophages toward an anti-tumor state, and inhibits MDSC migration. Clinical trials suggest low-dose metformin can reprogram the immune environment in esophageal cancer, showing potential as therapeutic option. ESCC, esophageal squamous cell carcinoma; AMPK, 5′ AMP-activated protein kinase; mTOR, the mammalian target of rapamycin protein; STAT3, signal transducer and activator of transcription 3; TNF-a, tumor necrosis factor alpha; IFN-γ, interferon gamma. Created with Biorender.com (accessed on 2 March 2024).

**Figure 4 ijms-25-02978-f004:**
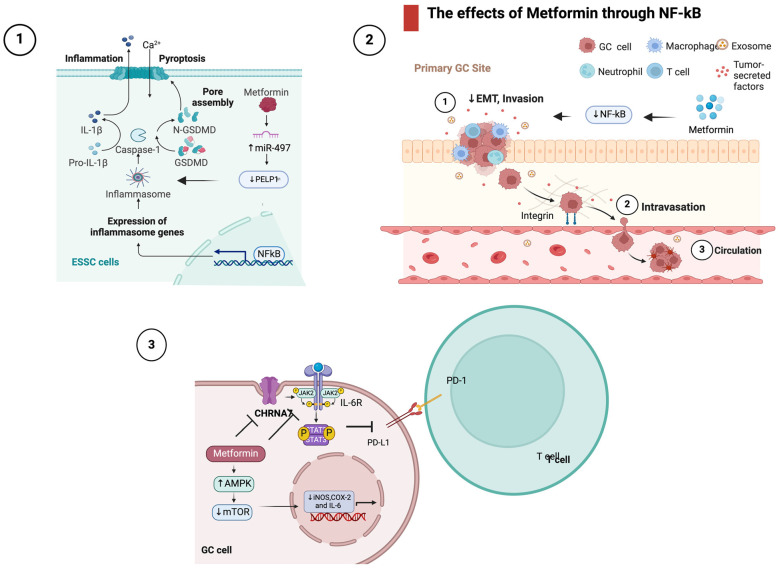
The metformin’s diverse impact on EC. Metformin demonstrates anti-inflammatory effects: it (1) induces pyroptosis and (2) inhibits NF-κB to impede EC progression. (3) It also disrupts CHRNA7, targeting the JAK2/STAT3/SOX2 pathway, and suppresses PD-L1 via the IL-6/JAK2/STAT3 pathway, enhancing the antitumor immune response. NF-kB, nuclear factor kappa-light-chain-enhancer of activated B cells; CHRNA7, neuronal acetylcholine receptor subunit alpha-7 gene, JAK2, Janus kinase 2; STAT3, signal transducer and activator of transcription 3 protein; SOX2, SRY (sex-determining region Y)-box 2 protein; PD-L1, programmed death-ligand 1 protein; IL6, interleukin 6; AMPK, 5′ AMP-activated protein kinase; mTOR, the mammalian target of rapamycin protein; IL-1β, interleukin 1β; GSDMD, gasdermin D protein; GC cell, germinal centers are transient structures that form within peripheral lymphoid organs in response to T cell-dependent antigen 1. Created with Biorender.com (accessed on 2 March 2024).

**Table 1 ijms-25-02978-t001:** Table summarizes the studies regarding EC risk reduction. Abbreviations: EC, esophageal cancer; ESCC, esophageal squamous cell carcinoma; EAC, esophageal adenocarcinoma; GC, gastric cancer; CRC, colorectal cancer; HCC, hepatocellular cancer; PC, pancreatic cancer; T2DM, type 2 diabetes mellitus; HR, hazard ratio; OR, odds ratio; CI, confidence interval.

Author/Year	Type of Cancer	Population	Concentration/Duration of Metformin Treatment	Results	Ref.
Lee et al. (2011)	EC, GC, CRC, HCC, PC	480,984 adult Taiwanese participants with T2DM vs. 417,844 non-DM controls	Mean metformin dosage was expressed in daily 500 mg units	↓ CRC and HCC incidences, depending on gender and cancer type (CRC in women, HCC in men), metformin HRs (95% CI): total 0.12 (0.08–0.19), CRC 0.36 (0.13–0.98), HCC 0.06 (0.02–0.16), PC 0.15 (0.03–0.79)], metformin dosage for a significant decrease in cancer incidence was ≤500 mg/day.	[33]
Tseng et al. (2017)	EC	288,013 metformin-treated T2DM Taiwanese adults vs. 16,216 other antidiabetic-drug-treated T2DM Taiwanese adults	Duration of metformin ≥ 2 years	↓ EC [HR (95% CI) 0.487 (0.347–0.684)]	[34]
Becker et al. (2013)	EC	All EC-T2DM patients in the GPRD (40–89 years of age, from1994–2010) vs. EC-free T2DM controls (up to 10 controls for each case)	Long-term (≥30 prescriptions) use	Not associated with a materially altered risk of esophageal cancer (adj. OR 1.23, 95% CI 0.92–1.65)	[35]
Wang et al. (2020)	ESCC Swedish	411,603 T2DM adults vs. 4,116,030 non-T2DM controls	Long-term or 1-year use	↓ ESCC [HR 0.68, 95% CI 0.54–0.85], especially in new-metformin users	[36]
Loomans-Kropp et al. (2021)	EAC	1943 EAC cases vs. 19,430 controls	≥2 prescriptions in the same drug category on different days and drug use must have occurred prior to study selection	Metformin use alone showed significant ↓ EAC risk among all participants [ OR 0.65; 95% CI 0.50, 0.82)] and those without BE [OR 0.99; 95% CI 0.28, 3.46]	[37]
Chen et al. (2020)	EC	Meta-analysis of 7 studies with 5,426,343 subjects	NA	↓ EC [HR = 0.69, 95% CI 0.54 to 0.87, *p* < 0.001]	[38]
Wu et al. (2020)	EC	Meta-analysis of 5 studies	NA	Metformin did not reduce EC risk in T2DM patients (HR 0.88, 95% CI 0.60–1.28, *p* > 0.05). Subgroup analyses by geographic location showed that metformin ↓ EC in Asian patients withT2DM (HR 0.59, 95% CI 0.39–0.91, *p* = 0.02), without heterogeneity between studies (*p* = 0.80 and I2 = 0%).	[39]

**Table 2 ijms-25-02978-t002:** The influence of metformin on survival rates of EC. Abbreviations: EC, esophageal cancer; ESCC, esophageal squamous cell carcinoma; EAC, esophageal adenocarcinoma; T2DM, type 2 diabetes mellitus; CRT, chemoradiation therapy; HR, hazard ratio; OR, odds ratio; CI, confidence interval; *n*, number of patients; CR, complete response rate; pCR, pooled complete response rate; NA, non-applicable.

Author/Year	Type of Cancer	Population	Concentration/Duration of Metformin Treatment	Results	Ref.
Wang et al. (2023)	EC	T2DM + no metformin (*n* = 379), no T2DM + no metformin (*n* = 3999), T2DM + metformin (*n* = 473)	Any dose	↓ all-cause mortality in non-T2DM patients and metformin-T2DM patients, ↓ HRs of all-cause mortality with a higher daily dose of metformin (Ptrend = 0.04)	[40]
Skinner et al. (2013)	EAC	286 EAC patients treated with concurrent CRT followed by esophagectomy (29 T2DM + metformin patients, 21 T2DM + no metformin patients, 235 non-T2DM)	Any dose	↑ CR rate in T2DM + metformin patients vs. T2DM + no metformin patients (34.5% vs. 4.8%, *p* = 0.01) and vs. non-T2DM (19.6%, *p* = 0.05), ↑ CR rate with ≥ 1500 mg/d metfromin, ↓ in field loco-regional failure following radiation (*p* = 0.05)	[41]
Spierings et al. (2015)	EC	461 EC patients treated with concurrent CRT followed by esophagectomy (32 T2DM + metformin patients)	Any dose	No differences in pathological response rates or overall survival or disease-free survival between meformin to non-metformin users	[42]
Voorde et al. (2015)	EC	196 EC adult patients (19 T2DM + metformin patients, 5 T2DM + no metformin patients, 172 non-T2DM)	Any dose	↑ distant metastasis-free survival rate (*p* = 0.040), ↑ overall survival rate (*p* = 0.012), ↑ survival (*p* = 0.043)	[43]
Sakamoto et al. (2022)	EC, rectal cancer	meta-analysis of 5 studies with 2041 patients	NA	↑ pCR rate (OR= 0.51 [0.34–0.76], *p* < 0.01), a positive correlation of metformin with EAC (coefficient = 0.13 [0.02–0.25], *p* = 0.03) and fluoropyrimidine anticancer drug use (coefficient = 0.01 [0.001–0.02], *p* = 0.03).	[44]

**Table 3 ijms-25-02978-t003:** A summary of Metformin’s role as a chemopreventive agent. Abbreviations: EC, esophageal cancer; ESCC, esophageal squamous cell carcinoma; EAC, esophageal adenocarcinoma; BE, Barrett’s esophagus; T2DM, type 2 diabetes mellitus; CRT, chemoradiation therapy; HR, hazard ratio; OR, odds ratio; aOR, adjusted odds ratio; CI, confidence interval; *n*, number of patients; CR, complete response rate; pCR, pooled complete response rate; NA, not applicable.

Author/Year	Type of Cancer	Population	Concentration/Duration of Metformin Treatment	Results	Ref.
Arai et al. (2022)	EC	308,793 patients (1911 ESCC, 195 EAC) and 306,687 non-EC patients	Any dose	↓ risk of ESCC (aOR 0.42, *p* < 0.0001)	[45]
Chak et al. (2015)	BE	74 subjects with BE	Randomly assigned to groups given metformin daily (increasing to 2000 mg/day by week 4, *n* = 38) or placebo (*n* = 36) for 12 weeks.	No differences in esophageal levels of pS6K1or epithelial proliferation or apoptosis in esophageal tissues.	[46]
Agrawal et al. (2014)	BE, EC	583 patients (115 EAC, 468 BE)	Any dose	No protective effect of metformin	[47]
Antonowicz et al. (2021)	EAC	Cell lines: FLO-1, OACM5.1, ESO26, KYAE-1, OE33, CPA, CPB, CPD	NA	↑ short-chain alkanals and medium-chain alkanals -> ↓ genotoxicity	[48]

**Table 4 ijms-25-02978-t004:** A comprehensive summary of the existing evidence regarding the impact of metformin on various preclinical models of EC. ESCC, esophageal squamous cell carcinoma; EAC, esophageal adenocarcinoma; p21, cyclin-dependent kinase inhibitor 1; p27, cyclin-dependent kinase inhibitor 1B; cyclin D1, regulatory subunit of cyclin-dependent kinases CDK4 and CDK6; mTOR, the mammalian target of rapamycin protein; USP7, ubiquitin-specific processing protease 7; Rb, retinoblastoma protein; EGFR/ErbB2/ErbB3, epidermal growth factor receptor; PYK, pyruvate kinase; VEGF, vascular endothelial growth factor; TIMP, metallopeptidase inhibitor 1.

Author/Year	Type of EC/Cell Line/Animal Model	In Vitro/In Vivo	Outcomes	Ref.
Xu et al. (2013)	ESCC/Eca-109, TE-1 cells	In Vitro	↑ cells in the G1/G0 phase ↓ cells in the S phase ↑ p21/p27 ↓ cyclin D1 expression activation of AMP/inhibition of mTOR signaling ↑ USP7 mRNA	[49]
Kobayashi et al. (2013)	ESCC/T.T, KYSE30,KYSE70	In Vitro	↑ cells in the G1/G0 phase ↓ cyclin D1/Cdk4/Cdk6/Rb expression	[50]
Damelin et al. (2014)	ESCC/WHCO1, WHCO5, SNO	In Vitro	↑ cells in the G1/G0 phase	[51]
Cai et al. (2015)	ESCC/EC109,EC9706/8-week-old male athymic nude mice	In Vitro/In Vivo	↑ cells in the G1/G0 phase, ↑ p53/p21/p27 ↓ cyclin D1 expression	[52]
Fujihara et al. (2015)	EAC/OE19, OE33, SK-GT4, OACM 5.1C/6-week-old male athymic nude mice	In Vitro/In Vivo	↑ cells in the G1/G0 phase ↓ cyclin D1/Cdk4/Cdk6/Rb expression ↓ *p*-EGFR/p-IGF-1R/ErbB2/ErbB3/insulin-R/PYK ↓ VEGF/TIMP-1/TIMP-2	[53]

**Table 5 ijms-25-02978-t005:** A comprehensive summary of key findings related to the impact of metformin on apoptosis in EC. ESCC, esophageal squamous cell carcinoma; EAC, esophageal adenocarcinoma; PARP, poly (ADP-ribose) polymerase; Bax/Bcl-2, apoptosis regulator BAX, also known as bcl-2-like protein 4; AVOs, acidic vesicular organelles; Beclin-1, mammalian ortholog of the yeast autophagy-related gene 6 (Atg6) and BEC-1 in the C. elegans nematode; p62, sequestosome-1 protein, which in humans is encoded by the SQSTM1 gene or the ubiquitin-binding protein; PI3K, phosphoinositide 3-kinase; AKT, RAC, (Rho family)-alpha serine/threonine-protein kinase; mTOR, the mammalian target of rapamycin protein; 4EBP1, eukaryotic translation initiation factor 4E-binding protein 1; S6K1, ribosomal protein S6 kinase beta-1; p21, cyclin-dependent kinase inhibitor 1; cyclin D1, regulatory subunit of cyclin-dependent kinases CDK4 and CDK6; STAT3, signal transducer and activator of transcription 3; AXL, tyrosine-protein kinase receptor UFO; ULK1, Unc-51-like autophagy-activating kinases 1; LC3B-II, microtubule-associated proteins 1A/1B light chain 3B; pH3, phospho-Histone H3; IHC, immunohistochemistry.

Author/Year	Type of EC/Cell Line/Animal Model	In Vitro/In Vivo	Outcomes	Ref.
Feng et al. (2014)	ESCC/EC109, EC9706/5–6-week-old female nude mice	In Vivo and In Vitro	Inhibition of ESCC cell growth, depolarization of the mitochondrial membrane, ↑ caspase-cleaved PARP/the ratio of Bax to Bcl-2 proteins, ↓ cellular proliferation represented by pH3 IHC ↑ AVOs, ↑ Beclin-1 ↓ p62 inhibition of Stat3/Bcl-2 pathway in an AMPK-dependent and -independent manner ↓ the growth of cultured ESCC cells -> ↓ tumor size and weight, inactivation of Stat3/Bcl-2 activity in vivo	[54]
Tang et al. (2017)	ESCC/Eca109, EC9706/6-week- old nude mice	In Vivo and In Vitro	↓ PI3K/AKT/mTOR pathway	[57]
Wang et al. (2017)	ESCC/KYSE450/4–6-week-old male nude mice	In Vivo and In Vitro	↓ 4EBP1/S6K1/p-4EBP1/p-S6K1	[58]
Li et al. (2017)	ESCC/KYSE520, KYSE140, KYSE410, KYSE30, KYSE150, KYSE510/4–5-week-old female nude mice	In Vivo and In Vitro	↑ cells in the G1/G0 phase, ↑ p21 ↓ cyclin D1 expression induction of mitochondrion-dependent apoptosis in ESCC cells, ↑ intracellular glutathione level/ROS	[59]
Shafaee et al. (2019)	ESCC/TE1, TE8, TE11	In Vitro	↑ p53 ↓ Bcl-2 expression	[56]
Peng et al. (2020)	ESCC/EC109	In Vitro	↓ of the growth of EC109 cells induction of apoptosis in ESCC cells ↓ phosphorylation of Stat3/ Bcl-2 pathway	[55]
Hong et al. (2022)	EAC/SK-GT4, FLO 1/4-week-old female mice	In Vivo and In Vitro	↓ AXL expression -> ↓ p-AMPK/p-ULK1/LC3B-II ↓ of tumor growth	[60]

**Table 6 ijms-25-02978-t006:** The intricate details of metformin-mediated epigenetic alterations in EC. ESCC, esophageal squamous cell carcinoma; EAC, esophageal adenocarcinoma; miRNAs, small non-coding RNA molecules; PELP1, proline-, glutamic acid- and leucine-rich protein 1; Hsp27, heat shock protein 27; AKT, RAC (Rho family)-alpha serine/threonine-protein kinase; HK2, hexokinase 2; KLF10, Krueppel-like factor 10.

Author/Year	Type of EC/Cell Line/Animal Model	In Vitro/In Vivo	Outcomes	Ref.
Kobayashi et al. (2013)	ESCC/KYSE30	In Vitro	↑ 17 miRNAs ↓ 45 miRNAs	[50]
Isozaki et al. (2014)	ESCC/TE2, T.Tn	In Vitro	Arrest of the G0/G1 and G2/M phases of the cell cycle, regulation of [67] the miR-375-targeted genes	[67]
Wang et al. (2019)	ESCC/KYSE510, KYSE140	In Vitro	Induction of human ESCC pyroptosis by targeting the miR-497/PELP1 axis	[69]
Liu et al. (2019)	ESCC/CE81T, TE1/6–8-week-old male NOD/SCID mice	In Vitro and In Vivo	↑ glycolysis and oxidative phosphorylation via the Hsp27–AKT–HK2 pathway	[70]
Chen et al. (2021)	ESCC/Te13, Te1, Eca109/BALB/c nude mice	In Vitro and In Vivo	↓ miR-340-5p in hypoxic exosomes -> ↑ the expression of KLF10 -> ↑ radiosensitivity	[68]
Fujihara et al. (2015)	EAC/OE19, OE33, SK-GT4, OACM 5.1C/6-week-old male athymic nude mice	In Vitro and In Vivo	↑ 3 miRNAs ↓ 10 miRNAs	[53]

**Table 7 ijms-25-02978-t007:** A comprehensive summary, consolidating all available evidence on the efficacy of metformin and other therapeutic modalities in the treatment of EC. ESCC, esophageal squamous cell carcinoma; EAC, esophageal adenocarcinoma; Bax/Bcl-2, apoptosis regulator BAX, also known as bcl-2-like protein 4; PI3K, phosphoinositide 3-kinase; AKT, RAC(Rho family)-alpha serine/threonine-protein kinase; AMPK, 5′ AMP-activated protein kinase; mTOR, the mammalian target of rapamycin protein; cyclin D1, regulatory subunit of cyclin-dependent kinases CDK4 and CDK6; CSCs, cancer stem cells; NAD(P)H, nicotinamide adenine dinucleotide phosphate; CDK4, cyclin-dependent kinase 4; HIF-1a, hypoxia-related factor-1α; TGF-β-Smad, transforming growth factor-β (TGF-β)-Smad phosphorylation pathway; IR, ionizing radiation; EMT, epithelial-to-mesenchymal transition; KLF10, Krueppel-like factor 10; DSF, disulfiram; Fbxo4, F-box protein 4; 3-ABA, 3-aminobenzamide; GLS, glutaminase.

Author/Year	Drug Combination	Type of EC/Cell Line/Animal Model	In Vitro/In Vivo	Outcomes	Ref.
Damelin et al. (2014)	cisplatin	ESCC/WHCO1, WHCO5, SNO	In Vitro	↑ glycolysis	[51]
				intracellular NAD(P)H levels + ↓ intracellular thiols -> ↓ cisplatin-DNA adduct formation	
Yu et al. (2016)	cisplatin	ESCC/ECA109	In Vitro	glucose-deprivation conditions -> ↑ metformin-associated cytotoxicity	[76]
				↓ cellular ATP levels	
				↓ AKT and AMPK signaling pathways -> ↓ DNA repair function -> ↑ cisplatin cytotoxicity	
Wang et al. (2017)	cisplatin	ESCC/KYSE450/4–6-week-old male nude mice	In Vivo and In Vitro	↑↑↑ anti-proliferation effects	[58]
Honjo et al. (2014)	5-fluorouracil (5-FU)	EAC/FLO-1, BE3,SKGT-4, OE33,JHESO, OACP/nude mice	In Vivo and In Vitro	↓ PI3K/mTOR signaling	[77]
				↓ CSCs -> inhibits EC cell proliferation	
Honjo et al. (2014)	5-fluorouracil (5-FU)	ESCC/YES-6, KATO-TN/nude mice	In Vivo and In Vitro	↓ PI3K/mTOR signaling	[77]
				↓ CSCs -> inhibits EC cell proliferation	
Feng et al. (2015)	IR	ESCC/ECa109	In Vitro	↑ cells undergoing G0/G1 cell cycle -> ↓ CDK4 and cyclin D1	[78]
				targeting the ATM and AMPK/mTOR/HIF-1a pathways -> ↓ HIF-1a -> ↑ radiosensitivity	
Nakayama et al. (2016)	IR	ESCC/TE-9	In Vitro	↓ TGF-β-Smad phosphorylation pathway, and a part of the non-Smad pathway -> suppression of IR-induced EMT	[79]
Chen et al. (2021)	IR	ESCC/Te13, Te1, Eca109/BALB/c nude mice	In Vivo and In Vitro	↓ miR-340-5p in hypoxic exosomes -> ↑ the expression of KLF10 -> ↑ radiosensitivity	[68]
Jivan et al. (2015)	disulfiram (DSF), copper-DSF(Cu-DSF)	ESCC/WHCO1, WHCO5, SNO	In Vitro	↑ copper transport -> ↑ DSF cytotoxicity	[80]
Qie et al. (2019)	CB-839 (glutaminase 1 inhibitor)	ESCC/TE1, TE7, TE8, TE10, TE15/4-week-old male athymic mice	In Vivo and In Vitro	direct mutation, or loss of regulatory E3 ubiquitin ligase Fbxo4 -> ↑↑ cyclin D1 -> Gln addiction -> induction of apoptosis and suppression of cell proliferation	[85]
Shafaee et al. (2019)	2-deoxy-d-glucose (2DG)	ESCC/TE1, TE8, TE11	In Vitro	↑ p53 ↓ Bcl-2 expression	[56]
Wang et al. (2022)	3-aminobenzamide (3-ABA)	ESCC/KYSE450, TE-10/5-week old BALB/c male nude mice	In Vivo and In Vitro	↑ the suppressive effect of 3-ABA on ESCC cell growth	[86]
				↑ 51 proteins + ↓ 12proteins localised within the nucleus/cytoplasm/extracellular space -> ↓ growth of ESSC cells/↓ invasiveness	

## Data Availability

Not applicable.

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
