# Peer review of "Metformin in Esophageal Carcinoma: Exploring Molecular Mechanisms and Therapeutic Insights"

_ijms, 2024, doi:10.3390/ijms25052978_

Round 1

Reviewer 1 Report

Comments and Suggestions for Authors

The manuscript proposed by Stavros P. Papadakos et al. aims to summarize knowledge on the role of metformin in esophageal cancer (EC) management. This topic is important because EC remains a formidable malignancy with still limited treatment options and high mortality rates. Based on the current knowledge the authors, at first, discussed metformin’s physiological functions and anticancer mechanisms. Then, they described the drug's clinical significance and therapeutic implications as well as presented its molecular mechanisms in EC. These parts are very interesting. Generally, the layout of the work is clear. The most important information was well summarized in figures and tables. The work includes most published current articles.

Minor remarks:

- I have concerns about the structure of the section 2. It contains a lot of numerical information and thus does make the flow of reading difficult. I would advise the authors to make a table with them.

- Ensure consistency in the text: “anticancer” or “anti-cancer”, “et al” or “et al.”, “in vitro” or “in vitro”, etc.?

- Correct the beginning of the sentence at lines 117-118. It should be “It was found…” or “They found…”.

- Title the tables appropriately.

- It may be worth preparing a list of abbreviations and symbols, at least in the figures and tables captions.

- There are some punctuation errors like double or missing spaces, e.g., in lines: 94, 139, 288, 289. 

Comments on the Quality of English Language

Generally, English is fine.

Author Response

First Department of Pathology

Laikon General Hospital

National and Kapodistrian University of Athens

February 29st, 2023

Dear Editor,

RE: Metformin in Esophageal Carcinoma: Exploring Molecular Mechanisms and Therapeutic Insights

We thank you and the Reviewers for carefully evaluating our manuscript and for their positive and constructive feedback. Hopefully, our responses below address the points raised by the reviewers. All changes made are presented in yellow in the revised manuscript in MS word.

Reviewer 1: The manuscript proposed by Stavros P. Papadakos et al. aims to summarize knowledge on the role of metformin in esophageal cancer (EC) management. This topic is important because EC remains a formidable malignancy with still limited treatment options and high mortality rates. Based on the current knowledge the authors, at first, discussed metformin’s physiological functions and anticancer mechanisms. Then, they described the drug's clinical significance and therapeutic implications as well as presented its molecular mechanisms in EC. These parts are very interesting. Generally, the layout of the work is clear. The most important information was well summarized in figures and tables. The work includes most published current articles.

Response: Dear Reviewer,

Thank you for your thoughtful review of our manuscript on metformin's role in esophageal cancer management. We appreciate your positive comments on the clear layout, effective use of figures, and comprehensive inclusion of current articles. Your insights have greatly improved the quality of our work.

Point 1: - I have concerns about the structure of the section 2. It contains a lot of numerical information and thus does make the flow of reading difficult. I would advise the authors to make a table with them.

Response:  Thank you for your insightful feedback on our manuscript. Your point about the numerical information in section 2 affecting the flow of reading is well taken. We completely agree with your suggestion and have made significant revisions.

In response to your recommendation, we have introduced three new tables to better organize and present the numerical data in section 2. We believe these additions enhance the clarity and readability of the manuscript.

Point 2: - Ensure consistency in the text: “anticancer” or “anti-cancer”, “et al” or “et al.”, “in vitro” or “in vitro”, etc.?

Response:  Thank you for raising your point. I want to inform you that your concern has been addressed.

Point 3: Correct the beginning of the sentence at lines 117-118. It should be “It was found…” or “They found…”.

Response:

In response to your comment, we have carefully considered your suggestion and have made the necessary revisions.

Point 4:  Title the tables appropriately.

Response:

In response to your comment, we have carefully considered your suggestion and have made the necessary revisions.

Point 5:  It may be worth preparing a list of abbreviations and symbols, at least in the figures and tables captions.

Response:

In response to your comment, we have carefully considered your suggestion and have made the necessary revisions.

Point 6: There are some punctuation errors like double or missing spaces, e.g., in lines: 94, 139, 288, 289.

Response:

In response to your comment, we have carefully considered your suggestion and have made the necessary revisions.

Reviewer 2:  I would like to recommend this review article for publication, but minor revision need to modify:

Response: Dear Reviewer,

Thank you for your positive feedback on our manuscript and for providing additional references that are relevant to the topic. We appreciate your valuable suggestions and have incorporated them into the revised version of the article.

We greatly appreciate your contribution, as these additional references and discussions further enrich the manuscript and provide a more comprehensive view of the topic. Thank you for bringing these references to our attention and we believe that the revised manuscript now better addresses these important aspects.

Point 1: Esophageal cancer is divided into esophageal squamous cell carcinoma and esophageal adenocarcinoma. Their causes and mechanisms of onset are different. Can metformin be effective in treating them, and are the mechanisms of treatment the same. I suggest that all of these contents can be introduced in this review.

Response: Dear Reviewer,

We would like to bring to your attention that we have introduced a new paragraph in the manuscript to address the issues raised in your previous feedback. Specifically, we have included a section discussing the division of esophageal cancer into esophageal squamous cell carcinoma (ESCC) and esophageal adenocarcinoma (EAC), highlighting the differences in their causes and mechanisms of onset. Furthermore, to enhance clarity, we have ensured that each table explicitly mentions whether the presented data are from EAC, ESCC, or both. This modification aims to provide a clear distinction and attribution of the data sources, contributing to the overall transparency and reliability of the information presented in the review.

We remain open to any further suggestions or revisions you may find necessary.

Thank you for your continued guidance and support.

Point 2: As a review, two figures are still not enough, it is recommended to add more figures The number of Tables is sufficient.

Response: Dear Reviewer,

I would like to express my gratitude for your insightful comments on the document. Your feedback has been invaluable. I have incorporated the suggestions, and I'm pleased to inform you that there are now four images in the document.

Reviewer 2 Report

Comments and Suggestions for Authors

I would like to recommend this review article for publication, but minor revision need to modify:

1. Esophageal cancer is divided into esophageal squamous cell carcinoma and esophageal adenocarcinoma. Their causes and mechanisms of onset are different. Can metformin be effective in treating them, and are the mechanisms of treatment the same. I suggest that all of these contents can be introduced in this review.

2. As a review, two figures are still not enough, it is recommended to add more figures The number of Tables is sufficient.

Author Response

(The authors gave the same response as above.)
